# Diagnosing and Correcting Concept Omission
# in Multimodal Diffusion Transformers

**Kanghyun Baek** [1]   **Jaihyun Lew** [1]   **Chaehun Shin** [2]   **Jungbeom Lee** [3†]   **Sungroh Yoon** [1 2 4 †]

## Abstract

Multimodal Diffusion Transformers (MM-DiTs) have achieved remarkable progress in text-to-image generation, yet they frequently suffer from concept omission, where specified objects or attributes fail to emerge in the generated image. By performing linear probing on text tokens, we demonstrate that text embeddings can distinguish a characteristic 'omission signal' representing the absence of target concepts. Leveraging this insight, we propose Omission Signal Intervention (OSI), which amplifies the omission signal to actively catalyze the generation of missing concepts. Comprehensive experiments on FLUX.1-Dev and SD3.5-Medium demonstrate that OSI significantly alleviates concept omission even in extreme scenarios. The code is available at https://github.com/KangHyun-dsail/OSI.

## 1. Introduction

Diffusion models (Ho et al., 2020; Dhariwal & Nichol, 2021; Rombach et al., 2022; Nichol et al., 2021; Saharia et al., 2022; Jun et al., 2026; Park et al., 2026) have driven rapid progress in image generation over the past few years. Building on these advances, text-to-image (T2I) models have increasingly adopted Multimodal Diffusion Transformers (MM-DiT) (Esser et al., 2024; Labs, 2024), which enhance prompt-image alignment through joint interaction between different modalities. Yet, despite these advances, these models still exhibit concept omission, a common failure where specific objects are entirely missing (object omission) or

their visual properties are ignored (attribute neglect).

Several studies (Xie et al., 2023; Li et al., 2023b; Jiang et al., 2024) have attempted to address these concept omission issues, yet they typically require additional training or incur significant computational costs during inference. Moreover, existing studies (Chefer et al., 2023; Rassin et al., 2023; Agarwal et al., 2023) have primarily focused on visual embeddings, leaving the role of text embeddings underexplored. While Chen et al. (2024) attempt to study the effect of text embeddings in object generation, their analysis was confined to the output of the CLIP text encoder (Radford et al., 2021), leaving it unclear how these embeddings are interpreted and utilized within a diffusion model.

In this paper, we first analyze concept omission from the perspective of text embeddings. We investigate whether the concept text tokens contain information regarding the presence of the concept during the generation process in MM-DiT. To this end, we employ linear probing (Alain & Bengio, 2016), training a classifier to detect concept omission in the embedding space. Specifically, we construct a dataset by collecting concept text token embeddings during generation, labeled based on the presence or absence of the concept in the generated images. By applying linear probing to each attention head, we observe that specific heads achieve high accuracy, indicating that they contain information that distinguishes omission from existence. As illustrated in Figure 3, the predicted probability of the probe remains low during early timesteps when the concept has not yet emerged, and rises as the concept begins to emerge. Based on this finding, we propose Omission Signal Intervention (OSI). By amplifying the omission signal activated when a concept is omitted, we increase the model's awareness of omission, thereby boosting the generation of the target concept. Specifically, we extract the direction corresponding to the omission signal from our constructed dataset and add this direction to the representation, which effectively reinforces the model's internal intent to generate the target concept.

To validate the effectiveness of OSI, we conduct comprehensive evaluations across two benchmark categories: object omission and attribute neglect. For object omission, we utilize a modified GenEval (Ghosh et al., 2023) dataset featuring prompts with two to six objects, alongside the non-

---

[1]Interdisciplinary Program in Artificial Intelligence, Seoul National University, Seoul, South Korea [2]Department of Electrical and Computer Engineering, Seoul National University, Seoul, South Korea [3]Department of Computer Science & Engineering, Korea University, Seoul, South Korea [4]AIIS, ASRI, INMC, and ISRC, Seoul National University, Seoul, South Korea. [†]Correspondence to: Jungbeom Lee <jbeomlee@korea.ac.kr>, Sungroh Yoon <sryoon@snu.ac.kr>.

*Proceedings of the 43ʳᵈ International Conference on Machine Learning*, Seoul, South Korea. PMLR 306, 2026. Copyright 2026 by the author(s).

**Object Omission**  **Attribute Neglect**

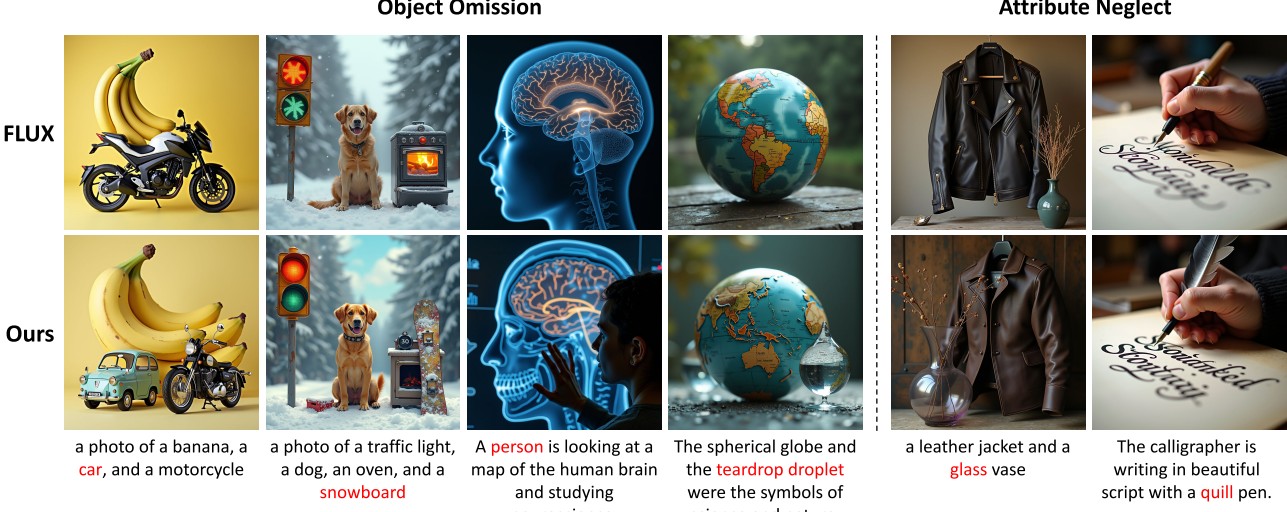

FLUX

Ours

| | | | | | |
a photo of a banana, a car, and a motorcycle | a photo of a traffic light, a dog, an oven, and a snowboard | A person is looking at a map of the human brain and studying neuroscience. | The spherical globe and the teardrop droplet were the symbols of science and nature. | a leather jacket and a glass vase | The calligrapher is writing in beautiful script with a quill pen.

*Figure 1.* Examples of concept omission: object omission (left) and attribute neglect (right). While the base model FLUX often fails to generate specific concepts within the prompt (highlighted in red), our method effectively mitigates such failures.

spatial subset of T2I-CompBench (Huang et al., 2023). For attribute neglect, we employ the attribute binding dataset from T2I-CompBench. Our experiments demonstrate that OSI achieves significant performance improvements across MM-DiT models: FLUX (Labs, 2024) and SD3.5 (stabilityai, 2024). These results strongly support our finding that the omission signal serves as a catalyst for concept generation.

Our key contributions are summarized as follows:

- We empirically verify that text tokens in MM-DiT capture omission-related information through joint interaction with image tokens.

- We introduce Omission Signal Intervention (OSI), which amplifies the omission signal to make the model aware of concept omission and boost generation of the target concept.

- We validate that OSI effectively relieves concept omission, yielding significant improvements across various scenarios.

## 2. Preliminaries

### 2.1. Flow Matching

Flow Matching (Lipman et al., 2023; Liu et al., 2023b) has emerged as a prominent generative framework, widely adopted by recent MM-DiT architectures (Esser et al., 2024; Labs, 2024; stabilityai, 2024). It defines a probability path $p_t$ which employs linear interpolation between a Gaussian

noise distribution $p_1$ and a data distribution $p_0$ as:

$$x_t = (1 - t)x_0 + tx_1, \tag{1}$$

where $x_0 \sim p_0$ and $x_1 \sim p_1$ for $t \in [0, 1]$.

From the probability path, a generation process is defined as an Ordinary Differential Equation (ODE):

$$\frac{dx_t}{dt} = v_t(x_t), \tag{2}$$

where the velocity field $v_t$ drives the probability flow from noise to data. The neural network $v_\theta$ is trained to predict the constant velocity $x_1 - x_0$ by minimizing the mean squared error.

During inference, samples are generated by solving the ODE using numerical solvers such as the Euler method, which iteratively updates the sample from $t = 1$ to $t = 0$:

$$x_{t-\Delta t} \approx x_t - \Delta t \cdot v_\theta(x_t, t). \tag{3}$$

Owing to the linear trajectory as in Equation 1, the clean data can be analytically estimated directly from any intermediate noisy state. Setting the step size $\Delta t = t$ in the numerical solver yields:

$$\hat{x}_0 \approx x_t - t \cdot v_\theta(x_t, t). \tag{4}$$

This facilitates the analysis of the generative trajectory by visualizing the corresponding clean targets. (Lv et al., 2025)

### 2.2. MM-DiT

Recently, the architecture of text-to-image (T2I) diffusion models has undergone a significant paradigm shift, transitioning from the conventional U-Net (Ronneberger et al.,

2015) framework to Multimodal Diffusion Transformers (MM-DiT) (Esser et al., 2024). While U-Net-based approaches rely on cross-attention mechanisms that inject textual information into visual features in a unidirectional manner, MM-DiT adopts a unified sequence processing strategy. In this architecture, text token embeddings and visual patch embeddings are concatenated into a single sequence, enabling joint attention across both modalities. The model applies multi-head self-attention by projecting the sequence into queries ($\mathbf{q}$), keys ($\mathbf{k}$), and values ($\mathbf{v}$) across multiple heads. Subsequently, the attention outputs from all heads are merged and added back to the residual stream, which carries the attended cross-modal information to the next layer. This suggests that the text embeddings inherently contain information regarding the current state of the generated image, serving as cues for subsequent layers to refine the generation. To simplify the notation in our subsequent analysis, we adopt the notation $\cdot^{(t,l,h)}$ to denote the components at head $h$ of layer $l$ at timestep $t$. (e.g., $\mathbf{k}^{(t,l,h)}$) When referring to the row vector of a specific text token $c$, we append the subscript $c$. (e.g., $\mathbf{k}_c^{(t,l,h)}$)

### 2.3. Attention Interpretation via Probing

Prior interpretability research across Large Language Models (meta llama, 2024) and Large Vision-Language Models (Liu et al., 2023a; Lee et al., 2024) has established that within multi-head attention mechanisms, specific heads specialize in capturing distinct information (Olsson et al., 2022; Li et al., 2023a; Yang et al., 2025). Motivated by these findings, we investigate whether specific heads within MM-DiTs capture information regarding the presence of target concepts through linear probing.

Probing is a diagnostic technique widely used to interpret the internal representations of neural networks (Li et al., 2023a; Liu et al., 2024; Alain & Bengio, 2016). This method involves training a lightweight classifier, a probe, on the frozen internal representations of a pre-trained model to predict a specific property associated with the input. Intuitively, the classification accuracy serves as a proxy for information accessibility: a high accuracy indicates that the information regarding the target property is encoded and readily extractable from the representation space.

## 3. Analysis on Concept Omission

To address the challenge of concept omission, we first focus our investigation on whether text token embeddings retain information regarding the successful generation or omission of its corresponding concept. To this end, we adopt the probing technique as described in Section 2.3. In our context, we train a linear classifier to distinguish between the 'presence' and 'absence' of a concept based on its text embeddings. We first proceed by detailing our dataset construction pipeline,

the collection of internal representations and labeling strategy for concept omission. (Section 3.1) Using the collected dataset, we analyze the internal representations of MM-DiT during the generation process. (Section 3.2) For experiments in this section, we use FLUX.1-Dev (Labs, 2024).

### 3.1. Dataset Construction for Probing

We construct a dataset $\mathcal{D}_{t,l,h} = \{(\mathbf{r}_i^{(t,l,h)}, y_i)\}_{i=1}^N$, where $\mathbf{r}_i^{(t,l,h)}$ denotes the internal representation of the concept token during generation at head $h$ of layer $l$ at timestep $t$ and $y_i \in \{0, 1\}$ denotes the presence label of the $i$-th sample. In the generation process, we utilize the two object subset from GenEval (Ghosh et al., 2023), which follows the template "a photo of obj1 and obj2". The presence labels are obtained from the final generation result, labeled as present ($y = 1$) if the concept is existent and absent ($y = 0$) if the concept is missing. We prioritize objects because attributes raise significant labeling challenges. Attributes are often entangled with objects, making it hard to isolate a missing attribute from objects. Yet, it is worth noting that our method successfully generalizes to attribute neglect (Section 5.2), although our analysis is mainly conducted on objects. We split this dataset into training and validation sets with a 4:1 ratio.

For our probing analysis, we choose key vectors $\mathbf{k}^{(t,l,h)}$ as the target internal representation $\mathbf{r}^{(t,l,h)}$, because key vectors of text tokens are known to have the greatest influence on image generation in MM-DiT models (Kim et al., 2025b; Shin et al., 2025). For labeling, we use Mask2Former (Cheng et al., 2022) from MMDetection (Chen et al., 2019) and BLIP-VQA (Li et al., 2022), and construct the probing dataset using instances where the two agree—i.e., both labelers predict the object to be present or absent.

**Filtering Noisy Pairs**   To prevent the probe from learning ambiguous patterns, we exclude the early and late time steps of the diffusion process from the training dataset, considering two key factors: the alignment between labels and internal representations, and the sufficiency of the signal encoded within those representations.

The exclusion of early timesteps is necessitated by the label misalignment; while the actual presence at these stages is mostly in an omission state, our labeling is tied to the final output. For example, the correct label for the early step should be 0 as the object does not exist yet; however, it could be assigned a label of 1 simply based on the final output. To quantify this misalignment, we measure the agreement between per-step $\hat{x}_0$ labels and the final image labels using the same labeling pipeline and timestep thresholds. As shown in Table 1, the agreement in the early timesteps is only 0.409, confirming that final image labels are poorly aligned with

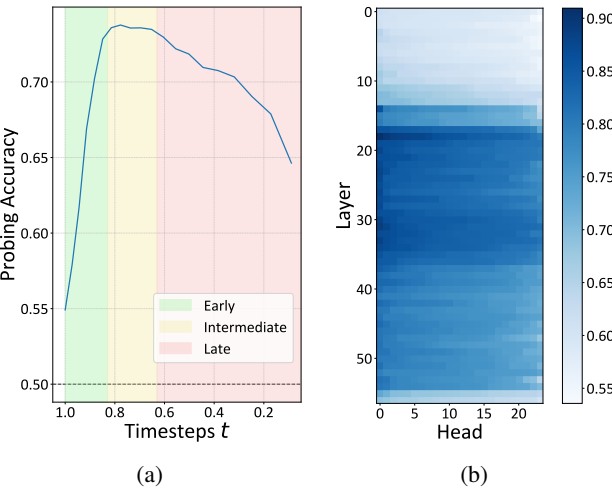

(a)        (b)

*Figure 2.* Analysis of probing accuracy across timesteps and heads. (a) Probing accuracy evaluated at each timestep, averaged across all heads. We observe that the accuracy peaks during the intermediate timesteps (yellow-shaded region), indicating that representations in this interval are most aligned with generation outcomes and contain sufficient information on omission. (b) Heatmap of head-wise probing accuracy averaged over intermediate timesteps. The results indicate that concept tokens in specific heads of the middle layers contain information that distinguishes between concept omission and existence.

*Table 1.* Agreement between per-step $\hat{x}_0$ labels and final image labels across timestep phases.

| Timesteps | Agreement with Final Label |
|---|---|
| **Early** | 0.409 |
| **Intermediate** | 0.965 |
| **Late** | 1.000 |

internal representations at this stage, whereas the agreement increases substantially in the later phases, reaching 0.965 in the intermediate timesteps.

We also exclude the late timesteps because they contain relatively little information on omissions, reflecting the coarse-to-fine nature of diffusion where late stages focus on fine-grained details rather than overall concept generation (Lv et al., 2025; Zhang et al., 2025; Choi et al., 2026). Figure 2a supports our claim. After training a classifier on data pairs from all timesteps, we observe that the average accuracy across all heads peaks during the intermediate timesteps (yellow-shaded region), while remaining relatively low in the early (green) and late (red) timesteps. Accordingly, we consider the data pairs from these intermediate timesteps $\mathcal{T}$ to be the most reliable pairs, and choose this subset $\mathcal{D}_{l,h}^{\text{train}} = \bigcup_{t \in \mathcal{T}} \mathcal{D}_{t,l,h}$ for probing.

### 3.2. Analyzing Internal Representations of Omission

For each curated dataset $\mathcal{D}_{l,h}^{\text{train}}$, we train a linear classifier and evaluate the probing accuracy. We visualize the probing accuracy for each head as a heatmap in Figure 2b, averaged across timesteps. We observe that probes trained in the early layers exhibit near-chance performance. We attribute this to the nature of MM-DiT architecture, where text tokens at these layers have not sufficiently incorporated information from image tokens yet. In contrast, probing performance improves substantially in the middle layers, reaching up to 91.0% accuracy in the top-performing head. This indicates that concept tokens contain information about omission status in a broad set of heads in middle layers. Therefore, we focus our omission analysis using top 300 attention heads with the highest accuracy out of the 1,368 total heads; all exceeding an accuracy of 80%.

**Temporal Evolution of Omission Signals**  We aim to observe how the probes could reflect the omission status throughout the generation process. Figure 3 visualizes the evolution of omission information across diffusion timesteps. The first row shows the predicted $\hat{x}_0$ at each timestep, while the second row summarizes the probe probabilities with respect to the concepts (*car*, *book*).

Figure 3 reveals a temporal pattern in probe responses. In early steps, visual concepts are absent in the predicted $\hat{x}_0$, and the predicted probabilities for both concept tokens are concentrated towards omission. As sampling progresses, $\hat{x}_0$ begins to exhibit recognizable instances of the concepts, and the probability distribution gradually shifts towards presence. Specifically, the visual features of the *book* begin to emerge in $\hat{x}_0$ at the fourth step, and those of the *car* in the fifth step. Correspondingly, their predicted probabilities also begin to rise starting from these respective steps. This demonstrates that the probe prediction is tightly synchronized with the visual emergence of the concepts.

Figure 4 demonstrates a quantitative evolution of this temporal behavior on the validation set, grouped by the presence labels. We observe that for both groups, the predicted probabilities are initially concentrated on absence. As generation progresses, the probabilities of the present group increase rapidly, whereas those of the absent group remain substantially lower. Overall, this quantitative trend aligns with our visual analysis in Figure 3: the omission signal is initially strong and diminishes in synchronization with the timeline of concept emergence.

## 4. Omission Signal Intervention

Leveraging the analysis from the previous section, we aim to mitigate concept omission. Our analysis reveals that the model contains an internal awareness of missing concepts

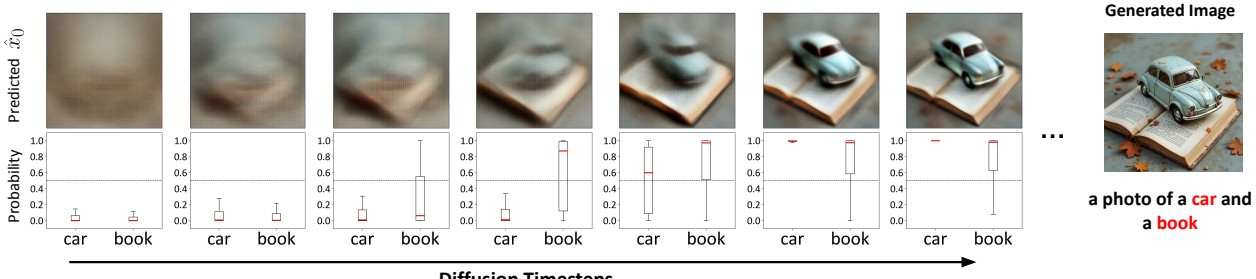

*Figure 3.* Visualization of concept emergence and corresponding probe predictions. The example is generated using the prompt "a photo of a car and a book". The top row displays the progression of the predicted image $\hat{x}_0$ across diffusion timesteps. The bottom row presents the distribution of probabilities for the corresponding concept tokens (*car*, *book*) with box plots. We confirm that as the concept starts to appear in the image, the probability increases accordingly.

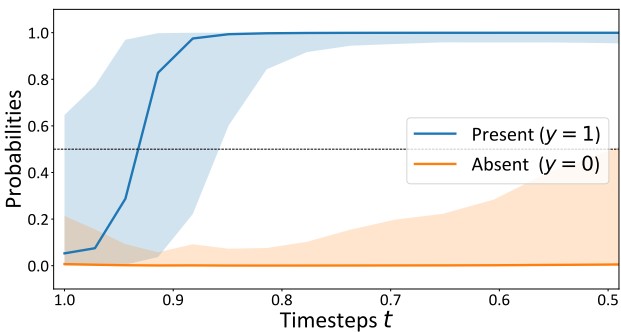

*Figure 4.* Temporal evolution of probe predictions on the validation set. The blue and orange colors correspond to probability distribution of each group labeled as present ($y = 1$) and missing ($y = 0$). The solid lines represent the median values aggregated over the selected top 300 heads, while the shaded regions indicate the interquartile range. (IQR) While the probability is concentrated on absence for both groups in early timesteps, the present group's probability increases as generation proceeds.

during the generation process. Consequently, we posit that by amplifying this 'omission signal,' we can induce a form of intentional hallucination regarding the severity of the missing concepts. By making the model perceive the omission as more critical than it is, we provide a powerful incentive that compels it to actively synthesize the missing objects.

Formally, using the curated dataset $\mathcal{D}_{l,h}^{\text{train}}$, we first compute the direction vector $\boldsymbol{\delta}^{(l,h)}$ defined by Mass Mean Shift (Li et al., 2023a), the difference between the mean representations of the omission ($y = 0$) and existence ($y = 1$) classes:

$$\boldsymbol{\delta}^{(l,h)} = \mathbb{E}[\mathbf{k}^{(t,l,h)} \mid y = 0] - \mathbb{E}[\mathbf{k}^{(t,l,h)} \mid y = 1]. \quad (5)$$

We then define the normalized steering direction $\boldsymbol{\theta}^{(l,h)}$:

$$\boldsymbol{\theta}^{(l,h)} = \frac{\boldsymbol{\delta}^{(l,h)}}{\|\boldsymbol{\delta}^{(l,h)}\|_2}. \quad (6)$$

Using this direction, we intervene on the key vectors $\mathbf{k}_c^{(t,l,h)}$ of a concept text token $c$ at layer $l$ and head $h$ according to the following linear shift:

$$\mathbf{k}_c^{(t,l,h)} \leftarrow \mathbf{k}_c^{(t,l,h)} + \alpha\sigma^{(l,h)}\boldsymbol{\theta}^{(l,h)}, \quad (7)$$

where $\sigma^{(l,h)}$ represents the standard deviation of the projections along $\boldsymbol{\theta}^{(l,h)}$ in the probing dataset, and $\alpha$ is a scalar hyperparameter controlling the intervention strength. We term this method Omission Signal Intervention (OSI), an approach that amplifies omission signal along the steered direction for alleviating concept omission.

However, excessive intervention may cause the model to deviate from the learned distribution. Therefore, to minimize unintended perturbations, we selectively apply intervention to top-$K$ heads with the highest probing accuracy. Furthermore, considering that concept formation primarily occurs in the early timesteps (Choi et al., 2022) of the generation process (starting from $t = 1$), we apply our intervention to specific time interval $t \in [t_{\text{stop}}, 1]$. For any head outside the top-$K$ or any timestep $t < t_{\text{stop}}$, we disable intervention by setting the steering vector $\boldsymbol{\theta}^{(l,h)}$ to zero, preserving the model's standard generation process.

## 5. Experiments

### 5.1. Implementation Details

**Experimental Settings** We conduct experiments on two MM-DiT models: FLUX.1-Dev (Labs, 2024) and SD3.5-Medium (stabilityai, 2024). For inference, all images are generated with total inference steps $T=30$. To ensure fair comparison, we use the default classifier-free guidance (CFG) (Ho & Salimans, 2022) scale for each model (3.5 for FLUX and 7.0 for SD3.5). We compare against two very recent and strong baselines, TACA (Lv et al., 2025) (LoRA rank $r=64$) and PLADIS (Kim & Sim, 2025). We evaluate PLADIS exclusively on FLUX due to implementation availability.

*Table 2.* Quantitative comparison on object omission. We evaluate models using GenEval (Ghosh et al., 2023) prompts with varying object counts (2–6), and the non-spatial subset of T2I-CompBench (Huang et al., 2023).

| Backbone | Method | GenEval | | | | | | T2I-CompBench |
|---|---|---|---|---|---|---|---|---|
| | | Two object | Three object | Four object | Five object | Six object | Avg. | Non-spatial |
| FLUX | Base | 0.81 | 0.63 | 0.44 | 0.29 | 0.18 | 0.47 | 0.3069 |
| | TACA | 0.89 | 0.68 | 0.56 | 0.31 | 0.20 | 0.53 | 0.3078 |
| | PLADIS | 0.87 | **0.71** | 0.56 | 0.31 | 0.20 | 0.53 | 0.3075 |
| | Ours | **0.92** | **0.71** | **0.64** | **0.40** | **0.40** | **0.61** | **0.3083** |
| SD3.5 | Base | 0.82 | 0.72 | 0.59 | 0.38 | 0.24 | 0.55 | 0.3155 |
| | TACA | 0.87 | 0.74 | 0.55 | **0.47** | 0.26 | 0.58 | **0.3164** |
| | Ours | **0.89** | **0.80** | **0.68** | 0.46 | **0.35** | **0.64** | 0.3159 |

*Table 3.* Quantitative comparison on attribute neglect. We evaluate models on the attribute binding subset of T2I-CompBench (Huang et al., 2023) across three categories (color, shape and texture).

| Backbone | Method | Color | Shape | Texture |
|---|---|---|---|---|
| FLUX | Base | 0.7923 | 0.4995 | 0.6419 |
| | TACA | 0.7742 | 0.5118 | 0.6493 |
| | PLADIS | 0.7914 | 0.5108 | 0.6441 |
| | Ours | **0.8014** | **0.5819** | **0.7039** |
| SD3.5 | Base | 0.7955 | 0.5820 | 0.7305 |
| | TACA | **0.8159** | 0.5948 | 0.7458 |
| | Ours | 0.8048 | **0.6119** | **0.7480** |

**Intervention Hyperparameters** For FLUX, we target the top-$K = 300$ heads out of total 1368 heads and set $K = 100$ out of 576 heads for SD3.5. We set the scaling coefficient $\alpha = 5.0$ and $\alpha = 7.5$ for FLUX and SD3.5 respectively, and our intervention is applied during the initial 15 steps of the total 30 inference steps for both models, which correspond to $t_{stop} = 0.78$ and $t_{stop} = 0.76$ each for FLUX and SD3.5.

**Evaluation Protocol** To evaluate object omission, we use the two-object prompts from GenEval (Ghosh et al., 2023) and construct additional multi-object prompts containing three to six objects (details provided in Appendix D.1). We further assess omission in action-centric scenarios using the non-spatial subset of T2I-CompBench (Huang et al., 2023), which includes prompts describing objects with specific verbs. To assess generalizability on attribute neglect, we use the attribute binding subset of T2I-CompBench. For this setting, we apply the intervention on both the target object tokens and their corresponding attribute tokens. For token selection, we adopt distinct approaches based on the prompt structure. Since GenEval and the custom multi-object prompts follow fixed templates, we use rule-based parsing. For T2I-CompBench, we employ Llama 3.1 8B Instruct (meta llama, 2024) to extract target object spans and associated attributes. The exact system instructions are provided in Appendix D.2.

## 5.2. Quantitative Results

We first evaluate the object omission performance of FLUX and SD3.5. As reported in Table 2, our method demonstrates superior performance across both model architectures. Specifically, on FLUX, our method achieves the highest accuracy across all benchmarks, surpassing both TACA and PLADIS. This efficacy extends to SD3.5, where our method consistently outperforms the base model. The performance of base models degrades significantly as the object count increases in GenEval, and baselines exhibit limited efficacy in the challenging multi-object regime. On the other hand, our method demonstrates strong performance with excellent generalization to prompts with higher object counts (three to six objects) and the non-spatial subset of T2I-CompBench, which consists of unseen objects.

We next evaluate attribute neglect using the attribute binding subset of T2I-CompBench, as detailed in Table 3. Our method demonstrates consistent effectiveness across architectures by achieving the best performance on FLUX and consistently improving over the base model on SD3.5 across all categories. Notably, although our probes are trained using object-level omission labels and the intervention heads/directions are selected accordingly, we observe additional improvements when also applying the intervention to the corresponding attribute tokens. This suggests that the omission-related signal we identify is not specific to object tokens alone and can transfer to attribute realization. Additional results of applying the intervention separately to object tokens and attribute tokens are included in Appendix E.2, and results on image quality metrics are also reported in Appendix E.1.

## 5.3. Qualitative Results

Figure 5 shows qualitative comparisons with baseline methods. In the first two rows, the baselines frequently omit some of the requested objects, such as cell phone, toaster or TV. In contrast, our method consistently renders all objects specified in the prompt. Moreover, the benefits extend beyond simple object enumeration to action-centric and

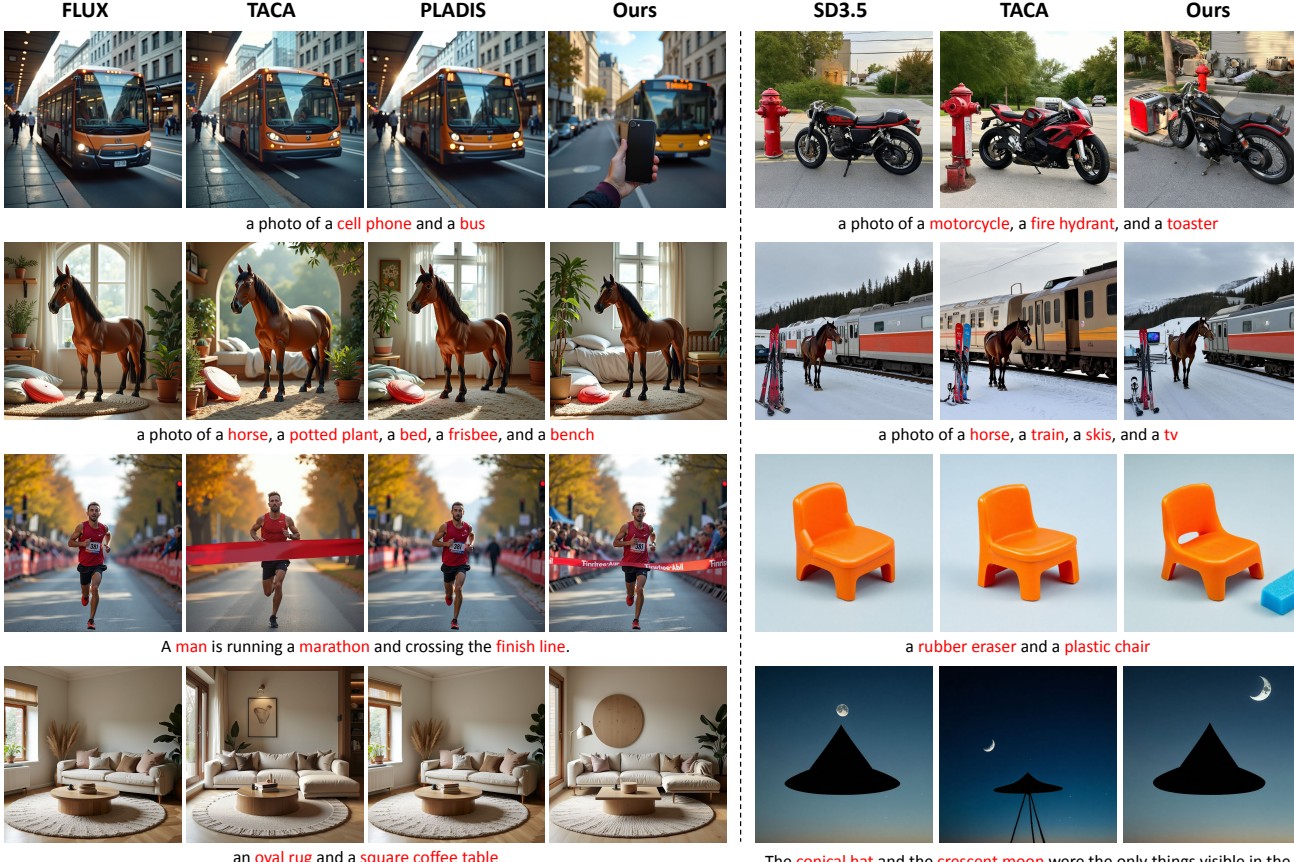

FLUX | TACA | PLADIS | Ours | SD3.5 | TACA | Ours

a photo of a cell phone and a bus

a photo of a horse, a potted plant, a bed, a frisbee, and a bench

A man is running a marathon and crossing the finish line.

an oval rug and a square coffee table

a photo of a motorcycle, a fire hydrant, and a toaster

a photo of a horse, a train, a skis, and a tv

a rubber eraser and a plastic chair

The conical hat and the crescent moon were the only things visible in the night sky

*Figure 5.* Qualitative comparison across various benchmarks. We present samples from the baselines based on FLUX and our method on the left, and the corresponding comparison for SD3.5 on the right.

attribute-specific prompts. For instance, in the third row of the left example, the baselines fail to capture the finish line while successfully depicting the runner. However, our method properly generates both. Similarly, in the last row, base models often ignore geometric constraints, generating a round table instead of a square one, or miss specific visual elements like the crescent moon. Our method corrects these cases, suggesting that it is effective not only for preventing object omission but also for resolving attribute neglect.

## 5.4. Ablation Study

We conduct ablation studies on the intervention direction and head selection strategy, based on FLUX. We first fix the intervention heads and ablate the direction in two ways: (i) the opposite direction $(-\boldsymbol{\theta}^{(l,h)})$, which reverses the omission signal, and (ii) a random direction sampled from a normal distribution. We then fix the intervention direction and ablate the head selection strategy by choosing (i) the bottom-$K = 300$ heads with the lowest probing accuracy, (ii) randomly selected $K = 300$ heads and (iii) all $K = 1368$ heads without exclusion.

*Table 4.* Ablations on intervention direction and head selection using FLUX. We use GenEval prompts with varying object counts (2–6). We compare our method against different intervention directions (Opposite, Random) and head selection strategies (Bottom, Random, All). We set $K = 300$ for all cases except 'All'.

| Setting | | Two | Three | Four | Five | Six |
|---|---|---|---|---|---|---|
| **Base (FLUX)** | | 0.81 | 0.63 | 0.44 | 0.29 | 0.18 |
| **Direction ($\theta$)** | Opposite | 0.72 | 0.40 | 0.15 | 0.06 | 0.02 |
| | Random | 0.84 | 0.65 | 0.53 | 0.32 | 0.30 |
| **Heads ($K$)** | Bottom | 0.81 | 0.60 | 0.47 | 0.22 | 0.15 |
| | Random | 0.82 | 0.67 | 0.55 | 0.31 | 0.17 |
| | All | 0.90 | **0.74** | 0.50 | 0.33 | 0.32 |
| **Ours** | | **0.92** | 0.71 | **0.64** | **0.40** | **0.40** |

**Intervention Direction** As shown in Table 4, intervening with the opposite direction generally degrades performance compared to the base model. This further confirms that our learned direction is semantically aligned with the object realization process; consequently, reversing this vector tends to disrupt the constructive features required for generation.

Conversely, while random directions yield marginal gains over the baseline, our selection of appropriate directions yields more substantial improvements.

**Intervention Head Selection** For head selection, choosing the bottom-ranked heads results in little to no improvement and sometimes a slight drop in performance, suggesting that these heads are unrelated to omission and thus have limited influence on object realization. Selecting heads randomly yields a small performance gain, likely because some informative heads are included by chance; however, it still underperforms our method, highlighting the importance of selecting top-ranked omission-related heads. Although intervention on all heads brings quite an improvement, our selection of top-$K$ heads display stronger results, implying the importance of choosing crucial heads. Our top-$K$ selection in OSI achieves the best performance, validating that top-ranked heads with high accuracy effectively capture omission signals and allow the OSI scheme to strategically amplify and exploit these signals for enhanced results.

**Token-Specific Intervention** While the direction ablation above confirms that our learned vector is meaningful for concept realization, it remains to be verified whether OSI establishes a causal link to concept generation or merely acts as a general useful steering direction. To address this, we conduct a token-specific intervention experiment. We curate 100 failure cases from the FLUX baseline where objects from a GenEval multi-object prompt were omitted, and regenerate the images under two conditions: applying OSI only to one of the omitted object tokens (OSI - Omitted) and applying OSI to the already present object tokens (OSI - Present). We also measure the probe probability of the omitted object token at the exact timestep the intervention concluded. The results are summarized in Table 5.

When OSI is applied to the omitted object token, the accuracy of the omitted object reaches 0.70 and the probe probability increases from 0.292 to 0.658. In contrast, when the intervention is applied to the already present tokens, the increases in both probability and accuracy are marginal. This contrast demonstrates that the effectiveness of OSI relies on being applied to the specific target token, confirming that our learned direction not only carries general steering benefits but also directly amplifies the specific token signal to compel concept generation.

**Hyperparameter Selection** We further investigate hyperparameters for intervention: the scaling coefficient $\alpha$ and the number of intervened heads $K$. We conduct a grid search on the GenEval two object subset and report both accuracy and image quality measured with MANIQA (Yang et al., 2022). As shown in the accuracy heatmap presented at the left of Figure 6, our method consistently outperforms the

*Table 5.* Token-specific intervention experiment on 100 failure cases from FLUX. OSI - Omitted applies the intervention to the omitted object token, while OSI - Present applies it to the already present object token.

| Method | Accuracy (Omitted Obj.) | Probe Prob. (Omitted Obj.) |
|---|---|---|
| **FLUX** | 0.00 | 0.292 |
| **OSI - Omitted** | 0.70 | 0.658 |
| **OSI - Present** | 0.14 | 0.298 |

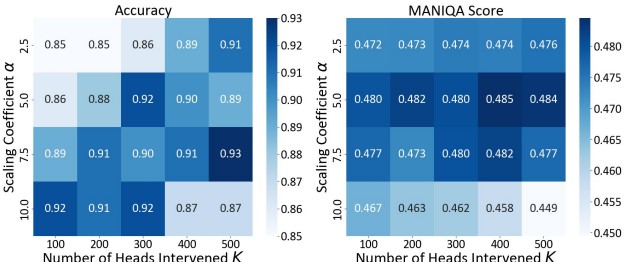

*Figure 6.* Ablation study on hyperparameters $K$ and $\alpha$. We report the accuracy (left) and MANIQA score (right) across different settings. Compared to the baseline FLUX (Accuracy: 0.82, MANIQA: 0.473), our method achieves robust improvements in alignment with minimal impact on image quality.

baseline across a wide range of configurations, demonstrating the robustness of our intervention. Notably, we find that moderate intervention strengths not only maintain but in some cases slightly improve MANIQA scores, confirming that our method is robust against quality degradation. However, since excessive intervention leads to perceptible quality drops, we select ($K$=300, $\alpha$=5.0) as the optimal configuration to balance alignment accuracy and perceptual fidelity.

**Intervention Interval** Finally, we examine the effect of the intervention duration $t_{\text{stop}}$, which defines the intervention window $[t_{\text{stop}}, 1]$. For clarity, Table 6 explicitly lists both the continuous $t_{\text{stop}}$ values and their corresponding discrete intervention timesteps. As presented in Table 6, applying the intervention even for the first 5 steps yields a significant boost in accuracy ($0.82 \rightarrow 0.88$), confirming that the early denoising phase is critical for determining object presence. We observe that the accuracy gain saturates at 15 steps. Thus we select 15 steps as the optimal stopping point to maximize performance with minimal intervention steps.

## 6. Related Work

Various approaches have been proposed to mitigate concept omission in text-to-image generation. One prominent stream employs inference-time guidance based on attention or constraint optimization (Chefer et al., 2023; Rassin et al., 2023; Agarwal et al., 2023; Xie et al., 2023) to explicitly enforce object generation or correct attribute binding. Another stream incorporates additional training to relieve concept

*Table 6.* Ablation on the intervention interval (number of steps). Results are reported on the GenEval two object subset using FLUX (30 total steps). Base denotes the model without intervention (0 steps).

| Step | 0 (Base) | 5 | 10 | 15 (Ours) | 20 | 25 | 30 |
|---|---|---|---|---|---|---|---|
| $t_{stop}$ | 1.00 | 0.95 | 0.87 | 0.78 | 0.64 | 0.44 | 0.00 |
| Accuracy | 0.82 | 0.88 | 0.91 | 0.92 | 0.92 | 0.92 | 0.91 |
| MANIQA | 0.473 | 0.479 | 0.480 | 0.480 | 0.481 | 0.480 | 0.480 |

omission, by learning box-grounding modules (Li et al., 2023b) or leveraging concept-matching feedback for alignment (Jiang et al., 2024). However, these methods often require fine-tuning the model or incur additional computational costs during inference. Additionally, while some works analyzed visual attention maps to address omission, they overlooked the text embeddings. Chen et al. (2024) identified that concept omission arises from mixed concept information in CLIP text embeddings. However, since their analysis was restricted to the text embeddings prior to entering the diffusion model, it was limited in revealing how these embeddings are processed within the model.

With the recent advancements in MM-DiT, research has increasingly focused on improving text-image alignment within this framework. PLADIS (Kim & Sim, 2025) improve prompt fidelity while modifying the attention mechanism using sparse attention. TACA (Lv et al., 2025) notes that text tokens fail to interact effectively due to the disparity in token counts, and proposes a rebalancing mechanism to address this. Seg4Diff (Kim et al., 2025a) identifies specific layers where textual tokens exert a strong influence on visual representations, and then fine-tunes the model to amplify these interactions for better alignment. While these methods improve overall prompt-image alignment, they still fail to solve concept omission. We investigate what specific information is encoded and transferred in the context of concept omission.

## 7. Limitations

While OSI effectively alleviates concept omission in a training-free manner, our method has several limitations. First, our intervention is sufficiently strong that it occasionally leads to over-generation, which can be undesirable depending on the context (see Figure 7). This side effect can be mitigated by adjusting hyperparameters to tune the intervention strength. Second, since OSI fundamentally relies on a clean, binary omission signal for training, its applicability to highly subjective or ambiguous concepts, such as aesthetic qualities or relative quantities, remains unexplored. Addressing these challenges in future work will allow us to further explore the scalability and universality of our approach.

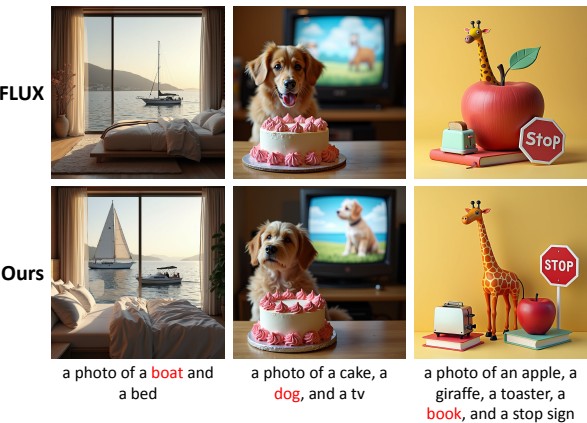

*Figure 7.* Examples of over-generation. Our method occasionally generates more instances of certain concepts (highlighted in red) compared to the FLUX baseline.

## 8. Conclusions

In this paper, we address the persistent problem of concept omission in Multimodal Diffusion Transformers. We identified that the concept text tokens have cues related to omission state and that these cues emerge in certain layers and heads at intermediate timesteps of generation. Based on these findings, we propose Omission Signal Intervention, a method which alleviates concept omission. Our method outperforms existing baselines in object omission benchmarks while effectively generalizing to attribute neglect scenarios. We expect this study to offer a new perspective on the mechanism of text-to-image generation.

## Impact Statement

This paper aims to advance the field of multimodal generative models by mitigating concept omission. Our proposed Omission Signal Intervention (OSI) enhances the fidelity of image generation without the computational cost of fine-tuning, thereby promoting more efficient and accessible tools for content generation. While improved generation capabilities could be misused to create misleading content, our method does not introduce new knowledge into the model but rather steers existing representations. We believe that the societal benefits of improved model controllability and efficiency outweigh the potential risks, provided that standard safety protocols for generative models are maintained.

## Acknowledgements

This work was supported by Institute of Information & communications Technology Planning & Evaluation (IITP) grant funded by the Korea government (MSIT) [No. RS-2021-II211343; RS-2022-II220959; Artificial Intelligence Graduate School Program (Seoul National University)]; the

National Research Foundation of Korea (NRF) grant funded by MSIT [No. 2022R1A3B1077720; 2022R1A5A7083908] and the BK21 Four program of the Education and Research Program for Future ICT Pioneers, SNU in 2026. This research was also conducted as part of the Sovereign AI Foundation Model Project (Data Track), organized by MSIT and supported by the National Information Society Agency (NIA) of Korea [No. 2025-AI Data-wi43]. This work was also supported by the National Research Foundation of Korea(NRF)[RS-2026-25488668] and Institute of Information & communications Technology Planning & Evaluation(IITP) under the artificial intelligence star fellowship support program to nurture the best talents [IITP-2026-RS-2025-02304828] grant, and IITP-ICT Creative Consilience Program grant [IITP-2026-RS-2020-II201819] funded by the Korea government(MSIT).

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

# A. Details of Probing Dataset Construction

**Prompt Construction.**   To ensure sufficient diversity for training the probes, we expand the dataset beyond the fixed evaluation set provided by GenEval (Ghosh et al., 2023). While we strictly adhere to the GenEval two-object template ("a photo of `obj1` and `obj2`"), we construct a broader set of prompts by randomly pairing object categories sampled from the MS-COCO (Lin et al., 2014).

**Labeling Strategy.**   To ensure reliable ground truth labels regarding concept presence, we employ a dual-verification strategy using Mask2Former (Cheng et al., 2022) from MMDetection (Chen et al., 2019) and BLIP-VQA (Li et al., 2022). Following GenEval (Ghosh et al., 2023), we apply the same confidence thresholds when using Mask2Former (Cheng et al., 2022). For BLIP-VQA, we query the model with the prompt "`object` ?" and compute the probability of the answer yes. We assign a positive label (1) if the probability exceeds 0.7, and a negative label (0) if it falls below 0.3. Instances where the probability falls between these thresholds are discarded to filter out ambiguous cases. Finally, as mentioned in the main text, we include only the samples where both models reach a consensus on the presence or absence of the concept.

**Token Aggregation.**   Since a single concept often consists of multiple tokens, we need a unified representation for probing. To address this, we average the key vectors of the constituent tokens to obtain a single vector representation per concept. This ensures that the input dimensionality for the linear probe remains consistent regardless of the token length of the concept.

**Dataset Balancing.**   A critical aspect of our dataset construction is the mitigation of bias on two levels. First, to prevent the probe from learning semantic identities rather than the omission signal, we enforce strict object-wise balancing. For every unique object, we ensure an exact match between the number of positive (present) and negative (absent) samples. Second, to prevent specific objects from dominating the dataset, we cap the number of samples per object at a maximum of 20. This ensures that the model's performance is not skewed towards a few frequently appearing concepts.

**Statistics.**   Through this pipeline, we construct a total of 822 samples derived from a candidate pool of 80 object categories. Specifically, 71 unique objects are represented in the final dataset with at least one sample. These are randomly partitioned into a training set of 654 samples and a validation set of 168 samples.

# B. Analysis on SD3.5-Medium

We extend our analysis to SD3.5-Medium (stabilityai, 2024) in addition to FLUX.1-Dev (Labs, 2024). Following the same protocol employed for FLUX, we construct a dataset comprising 654 samples across 67 unique objects. We partition this dataset into a training set of 516 samples and a validation set of 138 samples.

Unlike FLUX, which utilizes the output of the T5 text encoder (Raffel et al., 2020) exclusively as text tokens, SD3.5 employs outputs from both CLIP (Radford et al., 2021) and T5 encoders as text tokens. Consequently, we collect concept text token embeddings from both encoders and train separate classifiers for each to investigate their respective roles. The results are presented in Figure A1.

Figure A1a illustrates the probing accuracy evaluated at each timestep. We observe that the accuracy of the classifier trained on T5 tokens peaks during the intermediate timesteps (yellow-shaded region), exhibiting a trend analogous to the result of FLUX in Section 3.1. Conversely, the accuracy derived from CLIP tokens remains consistently low across all timesteps.

The head-wise heatmap analysis further corroborates these findings; while T5 A1b displays high accuracy patterns similar to FLUX, CLIP A1c exhibits low accuracy across nearly all attention heads. This empirical evidence suggests that the T5 text embeddings predominantly contain the discriminative information required to determine the omission status within the SD3.5 architecture.

Similar to Section 3.2, we selected 100 heads with probing accuracy exceeding 77% from the T5 probes to analyze the temporal evolution of omission signals. The results are presented in Appendix C.

# C. Head-wise Analysis of Omission Signals

**FLUX.1-Dev**   Figure A2 presents a fine-grained analysis of the aggregated temporal evolution shown in Figure 4 for FLUX, detailing the predicted probabilities for the selected 300 heads. To ensure readability while providing a comprehensive

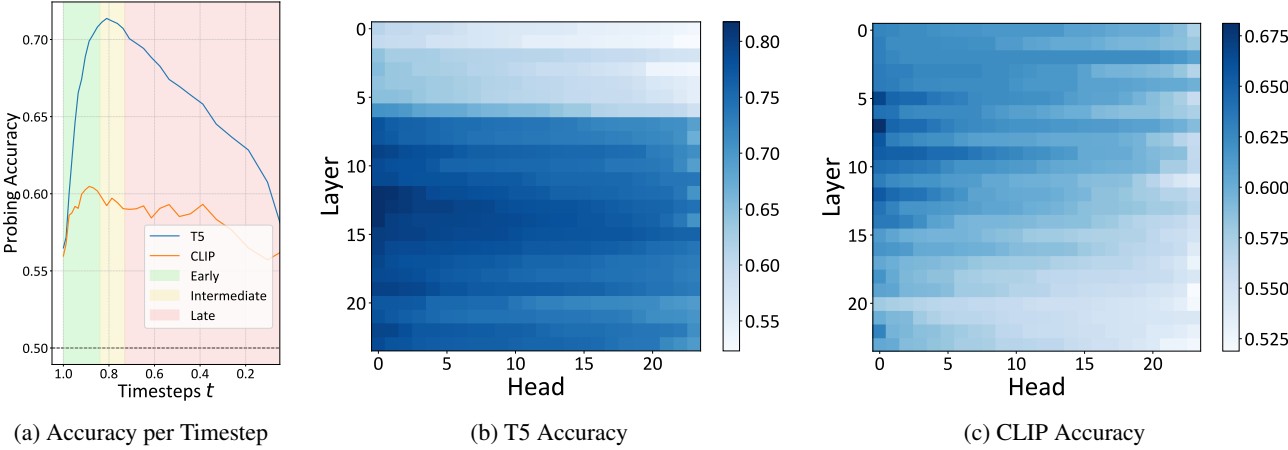

(a) Accuracy per Timestep        (b) T5 Accuracy        (c) CLIP Accuracy

*Figure A1.* Analysis of probing accuracy on SD3.5-Medium across timesteps and heads. (a) Probing accuracy evaluated at each timestep, averaged across all heads. We observe that the accuracy of the T5 encoder peaks during the intermediate timesteps (yellow-shaded region), whereas the accuracy of the CLIP encoder remains consistently low throughout the generation process. (b) Heatmap of head-wise probing accuracy for the T5 encoder, averaged over intermediate timesteps. Consistent with FLUX, specific heads in the middle layers exhibit high accuracy. (c) Heatmap of head-wise probing accuracy for the CLIP encoder, averaged over intermediate timesteps. In contrast to T5, the CLIP encoder demonstrates low accuracy across nearly all layers and heads.

view, we sort these heads by their probing accuracy and visualize them in batches of 10. Each subplot displays the averaged probabilities for each presence labels. We observe a consistent pattern across these top-performing heads: in the early timesteps, before the concept appears, the probabilities strongly indicate omission. As the diffusion process progresses, the predicted probability for the 'present' label gradually increases, effectively capturing the transition from absence to existence.

**SD3.5-Medium**    We apply the same methodology to visualize the temporal evolution for the top 100 heads of SD3.5, as presented in Figure A3. Similar to FLUX, we observe dominant omission signals in the early timesteps, regardless of the final label. However, considering the lower classifier accuracy and larger variance observed in Figure A3 compared to FLUX, we infer that SD3.5 possesses a relatively weaker capability to distinguish omission status information. We identify this limitation as the primary reason why the performance gains reported in Section 5.2 were not as significant as those for FLUX. Nevertheless, the fact that our method in SD3.5 still outperforms the baseline and exhibits a similar trend regarding omission signals suggests that our analysis is not limited to FLUX but captures a broader phenomenon.

## D. Prompt Construction and Extraction Details

### D.1. Construction of Multi-Object Prompts

To evaluate object omission in complex scenes, we constructed a dataset of prompts containing 3 to 6 objects. The construction process is as follows:

- **Vocabulary Source:** We utilized the object vocabulary from GenEval (Ghosh et al., 2023), which is derived from the standard **MS-COCO** (Lin et al., 2014) categories. This ensures that the prompts consist of common objects that the model is expected to recognize.
- **Random Sampling:** For a prompt with $N$ objects ($N \in \{3, \dots, 6\}$), we randomly sampled $N$ distinct objects from the vocabulary without replacement to ensure **no duplication** within a single prompt.
- **Template Structure:** Following the format of GenEval's two-object prompts, all prompts adhere to the template: "a photo of [Obj$_1$], [Obj$_2$], ..., and [Obj$_N$]."
- **Grammar:** Each object name is prefixed with the appropriate indefinite article ("a" or "an") based on its pronunciation. The objects are separated by commas, with an Oxford comma and the conjunction "and" preceding the final object.

Table A1 presents examples of the constructed prompts for different numbers of objects.

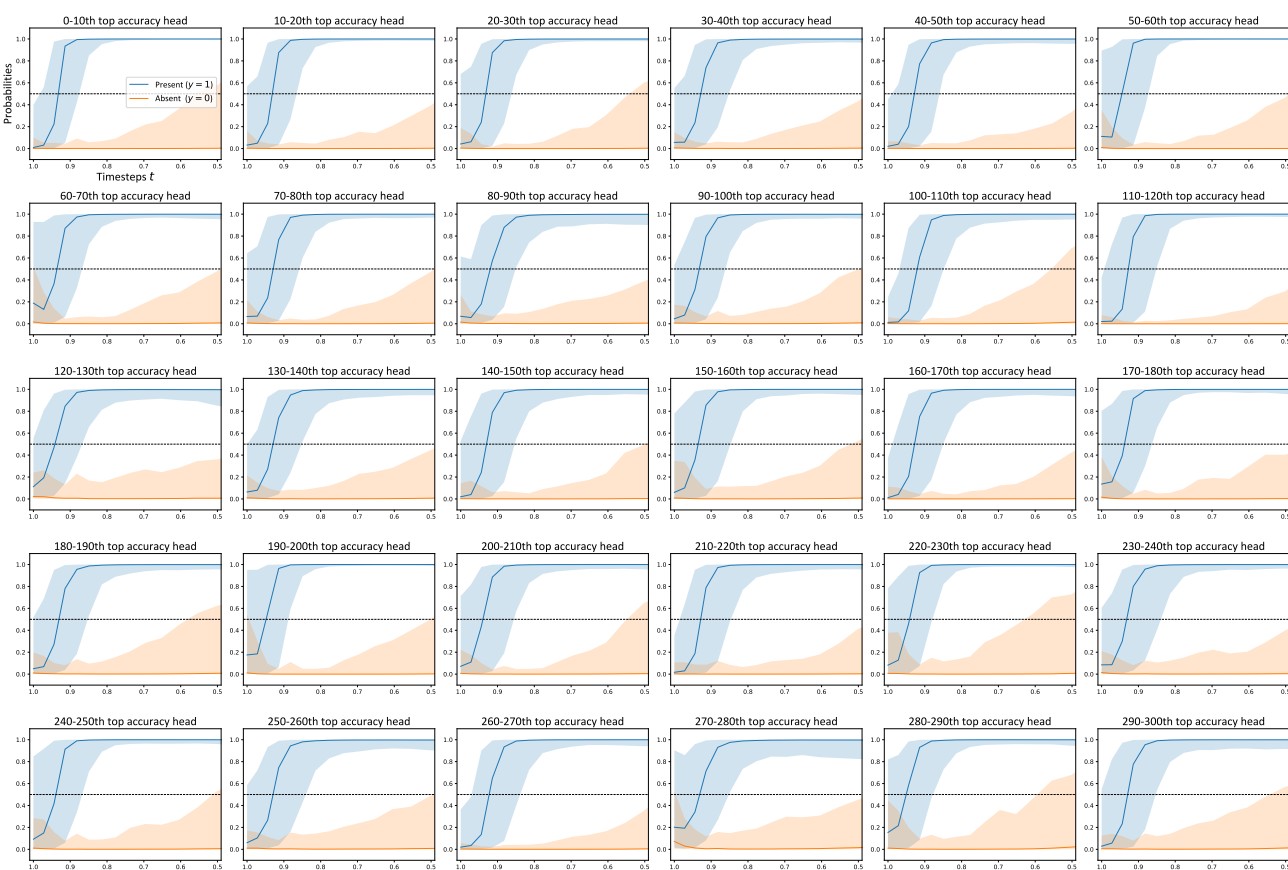

*Figure A2.* Detailed temporal evolution of predicted probabilities for the top 300 heads of FLUX.1-Dev. The heads are sorted by probing accuracy and visualized in batches of 10. For clarity, axis labels (Timesteps and Probabilities) are displayed only in the first subplot.

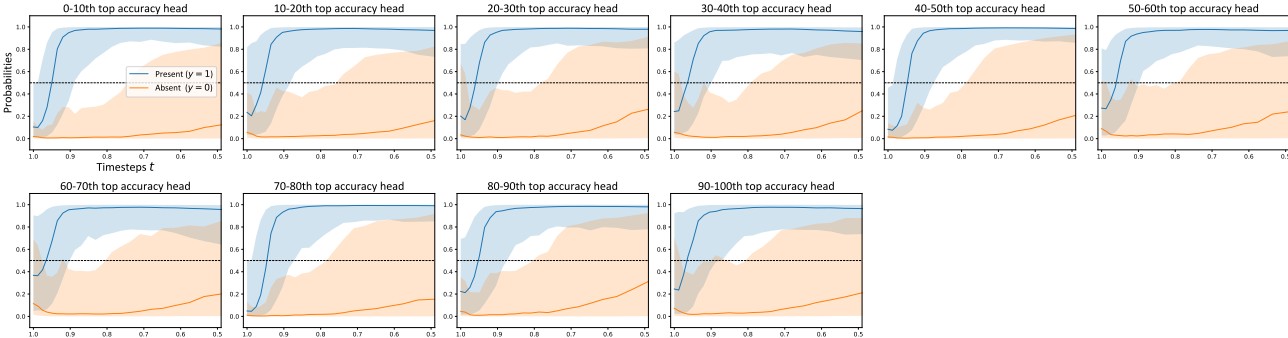

*Figure A3.* Detailed temporal evolution of predicted probabilities for the top 100 heads of SD3.5-Medium. Similar to Figure A2, the heads are sorted by accuracy and averaged in batches of 10.

## D.2. System Instructions for Extraction

To accurately identify target tokens for our analysis, we utilized Llama-3.1-8B-Instruct with specific system prompts tailored to the characteristics of each dataset subset. For the **T2I-CompBench Non-spatial subset**, which primarily evaluates object existence and action-object relationships, we focused on extracting distinct physical objects while strictly excluding visual attributes. The system prompt used for this task is presented in **Prompt D.2**. Conversely, for the **Attribute Binding subset**, assessing whether visual properties are correctly bound to objects is essential. Therefore, we designed the extraction strategy to capture objects together with their modifying adjectives as cohesive noun phrases, as detailed in **Prompt D.2**.

*Table A1.* Examples of constructed multi-object prompts.

| # Objects | Prompt Example |
|---|---|
| 3 | a photo of a cat, a dog, and a frisbee. |
| 4 | a photo of an apple, a horse, a backpack, and a dog. |
| 5 | a photo of a car, a truck, a bus, a traffic light, and a stop sign. |
| 6 | a photo of a cup, a fork, a knife, a spoon, a bowl, and a bottle. |

---

**System Prompt for Object Extraction**

You are an AI assistant that extracts visible, physical objects from text-to-image prompts.
Follow these rules strictly:
1. The extracted word must exist **exactly as written** in the original prompt.
   - Maintain Capitalization: If the prompt says 'Apple', extract 'Apple', not 'apple'.
   - Maintain Plurality: If the prompt says 'cars', extract 'cars', not 'car'.
2. Do NOT include **visual attributes (e.g., colors, textures, sizes, or shapes)**, **actions**, or **emotions**.
3. Output ONLY a Python list format. Do not add any explanation.

**Examples:**
```
Input: "A student is typing on a laptop at a coffee shop."
Output: ["student", "laptop", "coffee shop"]

Input: "A red apple is sitting on a wooden table."
Output: ["apple", "table"]
```

---

**System Prompt for Attributed Object Extraction**

You are an AI assistant that extracts descriptive noun phrases from text-to-image prompts.
Follow these rules strictly:
1. The extracted word must exist **exactly as written** in the original prompt.
   - Every extracted item must exist **exactly as a continuous string** in the original text.
   - *Example:* "The soft, fluffy texture of the cotton candy" → Extract "soft, fluffy texture", "cotton candy" (CORRECT)
   - *Example:* "The soft, fluffy texture of the cotton candy" → Extract "soft, fluffy cotton candy" (WRONG - not continuous)
   - Do NOT add words that are implied but not written.
   - Maintain Capitalization and Plurality exactly.
2. **Include Attributes (Adjectives)**:
   - Extract the object **together with its descriptive adjectives** (e.g., colors, textures, sizes, shapes).
   - Combine consecutive adjectives and the noun into a single string.
   - *Example:* "large red apple" (Include both "large" and "red").
3. **Handling Implied Nouns**:
   - If a noun is implied but omitted (e.g., "whitish on the other"), **extract ONLY the visible adjective**. Do NOT reconstruct the missing noun.
   - *Example:* "whitish on the other" → Extract "whitish" (CORRECT) vs. "whitish counter top" (WRONG).
4. Output ONLY a Python list format. Do not add any explanation.

**Examples:**
```
Input: "Person is looking at a waterfall and feeling refreshed."
Output: ["Person", "waterfall"]

Input: "A large red apple is sitting on a wooden table."
Output: ["large red apple", "wooden table"]

Input: "Kitchen scene, grey counter top on one side, whitish on the other and black
sink with long neck faucet."
Output: ["Kitchen scene", "grey counter top", "whitish", "black sink", "long neck
faucet"]
```

*Table A2.* Image quality scores on GenEval and T2I-CompBench.

| Backbone | Method | GenEval | | T2I-CompBench | |
|---|---|---|---|---|---|
| | | MUSIQ ↑ | MANIQA ↑ | MUSIQ ↑ | MANIQA ↑ |
| FLUX | Base | 0.7492 | 0.5435 | 0.7204 | 0.4832 |
| | TACA | 0.7481 | 0.5487 | 0.7202 | 0.4934 |
| | PLADIS | 0.7474 | 0.5446 | 0.7189 | 0.4870 |
| | Ours | 0.7535 | 0.5652 | 0.7190 | 0.4823 |
| SD3.5 | Base | 0.7558 | 0.5167 | 0.7280 | 0.4829 |
| | TACA | 0.7554 | 0.5267 | 0.7329 | 0.4939 |
| | Ours | 0.7538 | 0.5117 | 0.7264 | 0.4778 |

*Table A3.* Ablation study of intervention on object and attribute tokens.

| Backbone | Method | Color | Shape | Texture |
|---|---|---|---|---|
| FLUX | Base | 0.7923 | 0.4995 | 0.6419 |
| | Ours (object) | 0.7678 | 0.5206 | 0.6743 |
| | Ours (attribute) | 0.7885 | 0.5439 | 0.6898 |
| | Ours (all) | **0.8014** | **0.5819** | **0.7039** |
| SD3.5 | Base | 0.7955 | 0.5820 | 0.7305 |
| | Ours (object) | **0.8182** | 0.5940 | 0.7349 |
| | Ours (attribute) | 0.7923 | 0.5986 | 0.7437 |
| | Ours (all) | 0.8048 | **0.6119** | **0.7480** |

## E. Additional Results

### E.1. image quality assessment

Following TACA (Lv et al., 2025), we evaluate image quality using MUSIQ (Ke et al., 2021) and MANIQA (Yang et al., 2022) on images generated from GenEval (Ghosh et al., 2023) and T2I-CompBench (Huang et al., 2023) prompts, and report the averaged scores. As shown in Table A2, although our method intervenes at inference time in a training-free manner, it largely preserves image quality and yields scores comparable to the corresponding baselines across most settings. We attribute this to the fact that our intervention is localized to omission-relevant heads, which mitigates omission while minimizing degradation in overall image quality.

### E.2. Analysis of Intervention Targets for Attribute Neglect

We conduct an additional study on the T2I-CompBench attribute binding set to analyze how the choice of tokens for intervention affects performance. We separate tokens into object tokens and attribute tokens, and apply our intervention to each subset independently. Object tokens are extracted following Prompt D.2. Attribute tokens are defined as the remaining tokens after removing object tokens from the full set of extracted tokens. As shown in Table A3, intervening on either object tokens or attribute tokens alone improves performance in most cases. Importantly, intervening on both subsets jointly (Ours(all)) yields the strongest gains overall, suggesting that the omission-related signal we identify is not exclusive to object tokens and can transfer to attribute realization.

## F. Ablation on Intervention Interval and Steering Type

While our main experiments apply time-insensitive steering over the full interval $[t_{stop}, 1]$, two aspects of this design warrant further investigation: (1) whether there is a distribution mismatch between the early and intermediate timesteps since our probing dataset contains no samples from early timesteps, and (2) whether a time-dependent steering direction could better capture the evolving nature of features across timesteps. To address these, we conduct additional ablations combining two axes: the intervention interval ($[t_{stop}, 1]$ vs. $[t_{stop}, t_{early}]$) and the steering type (time-insensitive vs. time-dependent). The results are summarized in Table A5.

**Distribution Mismatch.** To investigate the extent of the potential distribution mismatch, we evaluated the average accuracy of our top 300 classifiers trained on intermediate timesteps when applied to early timestep features. As shown in Table A4, our classifiers maintain an accuracy of 0.772 even in the early timesteps, performing well above the chance level (0.500). While there is a slight performance drop compared to the intermediate timesteps (0.838), the fact that the classifiers retain such meaningful accuracy suggests that the distribution of the omission signal does not shift drastically between the early and intermediate timesteps. Furthermore, as shown in Table A5, restricting the intervention to $[t_{\text{stop}}, t_{\text{early}}]$ (Ablation 1) yields an accuracy of 0.83, falling short of our original performance (0.92). These results demonstrate that intervention during the early timesteps is highly effective in preventing concept omission, and that the benefits of early intervention outweigh the drawbacks of a marginal distribution mismatch.

**Time-Dependent Steering.** For time-dependent steering, we conduct two ablation studies: Ablation 2 trains time-dependent classifiers across the full $[t_{\text{stop}}, 1]$ interval, and Ablation 3 trains them within the $[t_{\text{stop}}, t_{\text{early}}]$ interval. Ablation 2 (0.87) yields lower performance compared to our time-insensitive method (0.92) over the same $[t_{\text{stop}}, 1]$ interval. We attribute this to the difficulty of assigning accurate labels at early timesteps, which leads to unreliable steering directions. When restricted to the intermediate training interval (Ablation 3), the time-dependent approach (0.84) performs similarly to the time-insensitive approach in Ablation 1 (0.83), demonstrating that the direction representing concept omission is relatively time-insensitive within our trained intermediate timesteps. Additionally, a time-dependent approach requires training and storing classifiers for every single timestep, and would need to be retrained if the number of inference steps changes. Therefore, our time-insensitive design is effective considering that it achieves strong performance while maintaining efficiency by combining the intermediate timesteps for training.

*Table A4.* Probe accuracy of classifiers trained on intermediate timesteps when evaluated on early and intermediate timestep features.

| Timesteps of Dataset | Probe Accuracy |
|---|---|
| **Early** | 0.772 |
| **Intermediate** | 0.838 |

*Table A5.* Ablation on intervention interval and steering type using FLUX. Results are reported on the GenEval two-object subset.

| Method | Interval | Steering Type | Two Obj. Acc. |
|---|---|---|---|
| **FLUX** | - | - | 0.81 |
| **Ours** | $[t_{\text{stop}}, 1]$ | Time-insensitive | 0.92 |
| **Ablation 1** | $[t_{\text{stop}}, t_{\text{early}}]$ | Time-insensitive | 0.83 |
| **Ablation 2** | $[t_{\text{stop}}, 1]$ | Time-dependent | 0.87 |
| **Ablation 3** | $[t_{\text{stop}}, t_{\text{early}}]$ | Time-dependent | 0.84 |

