# OpenReview forum: "Diagnosing and Correcting Concept Omission in Multimodal Diffusion Transformers"
_ICML.cc/2026/Conference — ICML 2026 regular_

### Official Review · Reviewer_TiFk · 2026-03-01

**Soundness:** 3
**Presentation:** 3
**Significance:** 3
**Originality:** 3
**Overall Recommendation:** 5
**Confidence:** 4

**Summary:**

The paper analyzes concept omission from the perspective of text embeddings in diffusion models by employing linear probing. It finds that text tokens in MM-DiT capture omission-related information through joint interaction with image tokens, and introduces an intervention that amplifies the omission signal to boost the generation of the target concept.

**Compliance With Llm Reviewing Policy:**

Affirmed.

**Final Justification:**

The rebuttal addressed my concerns. Adding the limitation section will make the paper stronger and I strongly suggest it.

**Key Questions For Authors:**

See weaknesses

**Limitations:**

The paper doesn't discuss limitations in the main paper. Acknowledging and presenting limitations can improve the paper.

**Strengths And Weaknesses:**

__Strengths__

The analysis that shows that the model contains an internal awareness of missing concepts during the generation process is convincing, and the idea to use it for generation intervention is compelling and original. The paper is well-written, and the ablation section and experiments provide useful empirical evidence for the steering success


__Weaknesses__

- The paper focuses on the analysis of object omission, and it is not clear if the same approach can generalize to more abstract concepts or object properties (color/texture).

- It is not clear how many examples are needed to be curated for good steering. How does the performance scale when the direction is computed with more/less examples?

---

> ### Author Rebuttal · Authors · 2026-03-31
>
> We thank the reviewer for finding our analysis convincing and our intervention method original. We appreciate the valuable feedback and address your concerns below.
>
> > **W1**: The paper focuses on the analysis of object omission, and it is not clear if the same approach can generalize to more abstract concepts or object properties (color/texture).
>
> To verify the effectiveness of our method when applied to attributes, we constructed a new dataset and trained a classifier specifically for colors. We generated images using prompts formatted as "a photo of a {color 1} {object 1} and a {color 2} {object 2}" and assigned presence labels for the colors. For labeling, we adopted the color evaluation criteria from GenEval and T2I-CompBench: we first obtained the object's bounding box using Mask2Former, and then identified the dominant color within that box using CLIP. Simultaneously, we utilized BLIP-VQA by querying "a {color} {object}?" and extracting the probability of the “yes” answer. We assigned a positive label (1) only when both labeling methods agreed that the specific colored object was present; otherwise, we assigned a negative label (0). After training the color classifier on this refined dataset, we conducted intervention experiments on the T2I-CompBench color benchmark. The results are summarized below:
>
> | Method | T2I-CompBench (Color) |
> | :--- | :---: |
> | **FLUX** | 0.7923 |
> | **Ours (Color Classifier)** | 0.8001 |
>
> Consistent with our findings using object classifiers, applying the intervention with the color classifier yielded performance improvements over the FLUX baseline. These results empirically demonstrate that our methodology generalizes to object attributes like color. However, as noted in our paper, defining robust labeling criteria for various attributes is challenging, and existing evaluation benchmarks (e.g., GenEval) are currently limited in their attribute coverage beyond color. Consequently, we focused our primary analysis on objects to ensure the most reliable and measurable results.
>
> > **W2**: It is not clear how many examples are needed to be curated for good steering. How does the performance scale when the direction is computed with more/less examples?
>
> To investigate the data efficiency and scalability of our approach, we evaluated how the number of training samples affects both the classifier’s performance and the resulting intervention accuracy. We varied the training set size ($N$) from 20 to 654 (all available data) and measured: (1) the classification accuracy of the top-300 feature heads, and (2) the GenEval two-object accuracy after OSI intervention. The results are summarized below:
>
> | $N$ (# of Samples) | 20 | 50 | 100 | 200 | 400 | all (654) |
> | :--- | :---: | :---: | :---: | :---: | :---: | :---: |
> | **Classifier Accuracy** | 0.765 | 0.823 | 0.825 | 0.839 | 0.841 | 0.843 |
> | **Two Object Accuracy** | 0.87 | 0.91 | 0.91 | 0.90 | 0.91 | 0.92 |
>
> The results show that our method achieves strong performance with only 50 samples. This indicates that a small number of examples is sufficient to find a reliable steering direction.
>
>
> > **Limitations:** The paper doesn't discuss limitations in the main paper. Acknowledging and presenting limitations can improve the paper.
>
> We appreciate the constructive feedback regarding the missing limitations section. To address this, we provide a detailed discussion on the limitations of our approach in terms of its failure modes and ability boundaries. We commit to adding this section to the final version of the paper.
>
> Regarding the failure modes, our intervention is sufficiently strong that it occasionally leads to over-generation, which can be undesirable depending on the context. According to ICML's author response policy, we provide visual examples of failure cases via an anonymous external link: [https://anonymous.4open.science/r/7C90/limitation.pdf](https://anonymous.4open.science/r/7C90/limitation.pdf). This side effect can be mitigated by adjusting hyperparameters to tune the intervention strength.
>
> Regarding the ability boundaries, while our method effectively applies to objects and objective attributes, its applicability to highly subjective or ambiguous concepts, such as aesthetic qualities (beautiful) or relative quantities (many), remains unexplored. Since OSI fundamentally relies on a clean, binary omission signal for training, extending it to these abstract categories requires establishing objective and robust labeling criteria to mitigate human or model bias.  Addressing these challenges in future work will allow us to further explore the scalability and universality of our approach.

---

> > ### Author Rebuttal · Reviewer_TiFk · 2026-04-02
> >
> > The paper addressed my concerns, and I will raise the score to Accept.

---

> > > ### Author Response · Authors · 2026-04-05
> > >
> > > We are glad that your concerns have been fully resolved. Thank you for your time throughout the review and discussion process.

---

### Official Review · Reviewer_oX3U · 2026-03-06

**Soundness:** 3
**Presentation:** 3
**Significance:** 3
**Originality:** 3
**Overall Recommendation:** 5
**Confidence:** 4

**Summary:**

This paper considers the problem of concept omission in Diffusion Models, particularly in MM-DiT architecture applied by popular FLUX.1-dev and SD3.5-Medium model. Through the lens of attention probing, the authors figured out the keys of text tokens capture essential information for concept emergence. Based on this insight, the authors introduce Omission Signal Intervention(OSI) to inject intentional hallucinations to emphasize the target concept during the diffusion process. Extensive experiments and ablations demonstrates that OSI effectively mitigates the concept omission in popular diffusion models.

**Compliance With Llm Reviewing Policy:**

Affirmed.

**Final Justification:**

The authors provided targeted additional experiments and clear explanations that directly addressed my original concerns, especially regarding the potential distribution mismatch and the time-insensitive steering design. The new evidence makes the key design choices much better justified, and I am satisfied that my concerns have been resolved. Overall, I would like to raise my score to 5.

**Key Questions For Authors:**

Same as weaknesses, i.e.:
1. Why is the intervention interval set to $[t_{stop}, 1]$ even if the curated dataset contains no sample for early timesteps?
2. Why is the steering direction time-insensitive?

**Limitations:**

Yes.

**Strengths And Weaknesses:**

## Strength
1. The analysis of concept omission in diffusion models (diagnosing) is convincing and effective, the probing results insightful.
2. The proposed method is effective, surpassing all the baselines, with generalization to attribute omissions which is more difficult.
3. Solid experiments and ablations are carried out, with thorough study on hyperparameters and distinctive attributes of different models, such as SD3's CLIP text encoder.
4. The overall presentation is clean and easy to understand.

## Weakness
1. Possible distribution mismatch in OSI. The paper states the steering interval being $[t_{stop}, 1]$, yet the dataset used in OSI ($D_{l,h}^{train}$) does not contain samples of early timesteps as mentioned in paragraph 'Filtering Noise Pairs' in sec 3.1. Will there be a mismatch? What will happen if the interval is modified to $[t_{stop}, t_{early}]$ where $t_{early}$ denotes the final early timestep?

2. Time-insensitive steering direction. Wide belief in the community (FreeControl[1], TACA[2], Lu et al.[3]) is that features within denoising network evolve as time progresses. Also, Fig. 2 demonstrates the correlation of text features to existance/absence is near-chance in early timesteps. The paper averages text features within all intermediate timesteps as steering direction for all time steps in $[t_{stop}, 1]$, contrary to common practice and findings in omission probing. This strategy needs further evidence such as comparison with time-dependent steering.

Despite these counter-intuitive designs, the overall performance achieved state-of-the-art, so I would recommend Weak Accept.

[1] FreeControl: Training-Free Spatial Control of Any Text-to-Image Diffusion Model with Any Condition

[2] Rethinking Cross-Modal Interaction in Multimodal Diffusion Transformers

[3] When Are Concepts Erased From Diffusion Models?

---

> ### Author Rebuttal · Authors · 2026-03-31
>
> We thank the reviewer for appreciating our convincing analysis and solid experiments. Following your constructive feedback, we conducted additional experiments. The results regarding the **distribution mismatch (W1)** and **time-dependent steering (W2)** are summarized below, followed by a detailed explanation for each concern.
>
> | Method | Intervention Interval | &nbsp; Steering Type | Two Object Accuracy |
> | :--- | :---: | :---: | :---: |
> | **FLUX** | - | - | 0.81 |
> | **Ours** | $[t_{\text{stop}}, 1]$ | Time-insensitive | 0.92 |
> | **Ablation 1 (W1)** | $[t_{\text{stop}}, t_{\text{early}}]$ | Time-insensitive | 0.83 |
> | **Ablation 2 (W2)** | $[t_{\text{stop}}, 1]$ | Time-dependent | 0.87 |
> | **Ablation 3 (W2)** | $[t_{\text{stop}}, t_{\text{early}}]$ | Time-dependent | 0.84 |
>
> > **W1**: Possible distribution mismatch in OSI. Why is the intervention interval set to $[t_{\text{stop}}, 1]$ even if the curated dataset contains no sample for early timesteps?
>
> As the reviewer insightfully noted, there may be a potential distribution mismatch between the early and intermediate timesteps. To investigate the extent of this mismatch, we conducted an additional experiment. At each denoising step, we directly predicted the clean image ($\hat{x}_0$) and assigned labels based on our manual assessment of whether the target object was present. We collected a dataset of 100 labeled samples for each of the early and intermediate timesteps. Using this dataset, we evaluated the average accuracy across our top 300 classifiers trained on intermediate timesteps. The results are summarized below:
>
> | Timesteps of Dataset |  &nbsp;&nbsp; Probe Accuracy |
> | :--- | :---: |
> | **Early** | 0.772 |
> | **Intermediate** | 0.838 |
>
> Our classifiers, although trained on intermediate timesteps, maintain an accuracy of 0.772 even in the early timesteps, performing well above the chance level (0.500). While there is a slight performance drop compared to the intermediate timesteps, we believe it is largely due to the difficulty of accurately judging object presence and assigning labels at early timesteps. Nevertheless, the fact that the classifiers retain such meaningful accuracy suggests that the distribution of the omission signal does not shift drastically between the early and intermediate timesteps.
>
> Furthermore, following your suggestion, we conducted the Ablation 1 experiment by restricting the intervention to the training interval $[t_{\text{stop}}, t_{\text{early}}]$ instead of starting from the initial timestep ($[t_{\text{stop}}, 1]$). As shown in the summary table above, Ablation 1 yields an accuracy of 0.83 on the GenEval two-object benchmark. While this is a marginal improvement over the FLUX baseline (0.81), it falls short of our original OSI performance (0.92). These results demonstrate that intervention during the early timesteps is highly effective in preventing concept omission. Since the benefits of early intervention outweigh the drawbacks of a marginal distribution mismatch, we designed our current approach.
>
> > **W2**: Why is the steering direction time-insensitive?
>
> Following your suggestion, we conducted two time-dependent ablation studies, detailed as Ablation 2 and 3 in the summary table above:
>
> - Ablation 2: Train time-dependent classifiers across the full $[t_{\text{stop}}, 1]$ interval and apply intervention at the corresponding timesteps.
> - Ablation 3: Train time-dependent classifiers within the $[t_{\text{stop}}, t_{\text{early}}]$ interval and apply intervention at the corresponding timesteps.
>
> Ablation 2 (0.87) yields lower performance compared to our time-insensitive method (0.92) over the same $[t_{\text{stop}}, 1]$ interval. To apply a time-dependent intervention starting from the initial timestep, we had to train separate classifiers for the early timesteps as well. However, attempting to train individual classifiers during these early steps, where it is difficult to assign accurate labels corresponding to the features, results in unreliable steering directions.
>
> Next, to verify the effectiveness of time-dependent classifiers specifically within our intermediate training interval, we analyzed Ablation 3. As shown, when restricted to this intermediate stage, the time-dependent approach in Ablation 3 (0.84) performs similarly to our time-insensitive approach in Ablation 1 (0.83). This demonstrates that even though features evolve across timesteps, the direction representing concept omission is relatively time-insensitive within our trained intermediate timesteps.
>
> Additionally, a time-dependent approach has the disadvantage of requiring us to train and store classifiers for every single timestep. We would also need to retrain them if we change the number of inference steps. Therefore, we believe that our time-insensitive design is effective considering that it achieves strong performance while maintaining efficiency by combining the intermediate timesteps for training.

---

> > ### Author Rebuttal · Reviewer_oX3U · 2026-04-04
> >
> > The rebuttal directly addresses my two main concerns with additional targeted experiments and clear explanations. For the possible distribution mismatch between early and intermediate timesteps, the authors provide both a probe-based analysis and an ablation that restricts intervention to the training interval. These results support the claim that early-timestep intervention remains effective despite some mismatch, and they clarify the design choice in a convincing way.
> >
> > The rebuttal also adequately addresses my concern about the time-insensitive steering direction. The newly added time-dependent ablations show that the proposed time-insensitive design performs better than, or at least as well as, the more fine-grained time-dependent alternatives considered here, while also being simpler and more efficient. Overall, the additional evidence makes the originally counter-intuitive design choices much better justified, and my main concerns are satisfactorily resolved.

---

> > > ### Author Response · Authors · 2026-04-05
> > >
> > > We are glad that our additional experiments and explanations have fully resolved your concerns. Thank you for your time throughout the review and discussion process.

---

### Official Review · Reviewer_S7Fs · 2026-03-13

**Soundness:** 3
**Presentation:** 2
**Significance:** 3
**Originality:** 3
**Overall Recommendation:** 4
**Confidence:** 3

**Summary:**

This paper studies concept omission in MM-DiT text-to-image models, including both object omission and attribute neglect. The main claim is that text token representations inside MM-DiT already contain an omission signal that reflects whether a target concept has appeared in the generated image. The authors diagnose this by applying linear probes to text-token key vectors across layers, heads, and timesteps, and show that some middle-layer heads can distinguish concept presence from absence with high accuracy. Based on this observation, they propose Omission Signal Intervention (OSI), which extracts an omission direction from the probing dataset and adds it to the concept token representation during inference, with intervention limited to selected heads and early denoising steps. Experiments on FLUX.1-Dev and SD3.5-Medium show improved performance on object omission and T2I benchmarks compared with recent baselines.

**Compliance With Llm Reviewing Policy:**

Affirmed.

**Final Justification:**

My concern has been basically resolved. Considering that I still do not have clear confidence in the ability boundaries of the method, I cannot raise the score to a clear acceptance. Therefore, my final score is weak accept.

**Key Questions For Authors:**

Please see the Weakness

**Limitations:**

The authors did not mention the limitations of the proposed method. The author should verify the capability boundary and failure mode of the proposed method by showing performance changes by task or giving more visualizations.

**Strengths And Weaknesses:**

#### Strength
1. The paper studies a clear problem. Concept omission is a common failure mode in modern MM-DiT text-to-image models, the paper focuses on a concrete setting propose solution.
2. The paper shows clear pipeline, which first use probing to show that text token contain omission-related information, then turns this analysis into a simple inference-time intervention solution.
3. The empirical results are reasonably strong on the targeted benchmarks. On object omission, the method improves over the base model and recent baselines, especially in harder multi-object settings.

#### Weakness
1. My main concern is that the paper’s mechanism claim is somewhat stronger than the evidence. The probe shows that omission-related information is linearly decodable from text-token representations, but this does not fully establish that the learned direction is the causal mechanism behind concept realization. The current results are consistent with the interpretation, but they do not rule out the possibility that OSI mainly acts as a useful steering direction rather than specifically amplifying a specific token signal.
2. The label construction for probing is not fully clean. Presence labels are assigned from the final generated image, while the internal representations come from intermediate timesteps. The authors try to address this by excluding early and late timesteps, which is reasonable, but this also highlights that the supervision is only approximately aligned with the intermediate representations.
3. Although the results demonstrate an average improvement in experience, as a training-free method, it is necessary to conduct a complete discussion of the ability boundaries and failure modes of this method.

---

> ### Author Rebuttal · Authors · 2026-03-31
>
> We thank the reviewer for appreciating our clear pipeline, from probing analysis to an intervention method, and for the valuable feedback. Below, we provide responses to address your concerns.
>
> > **W1**: Evidence does not fully establish the causal mechanism for concept realization, suggesting OSI might act as a useful steering direction rather than specifically amplifying a specific token signal.
>
> We provide additional evidence to support the causal interpretation between the learned direction and concept generation. First, we revisit our direction ablation study (Table 3),  which demonstrated that applying the opposite direction actively degrades performance, while a random direction yields only marginal improvements. This confirms that our learned vector is a meaningful direction for concept realization. Building on this, we conducted a token-specific intervention experiment to demonstrate that OSI is not merely a general useful direction, but also specifically amplifies the target token signal.
>
> Specifically, we curated a set of 100 failure cases from the FLUX baseline where some objects from a GenEval multi-object prompt were omitted. Using the exact same prompts and initial noise seeds, we regenerated the images under two distinct conditions: applying OSI only to one of the omitted object tokens (OSI - Omitted) and applying OSI to the already present object tokens (OSI - Present). We then measured the accuracy of the targeted missing object in both scenarios.
>
> Furthermore, since we showed in Figures 3 and 4 of our main paper that the probe predictions are tightly synchronized with the visual emergence of concepts, we also tracked the internal representational shift. Specifically, we measured the probe probability of the omitted object token at the exact timestep the intervention concluded. The results are summarized below:
>
> | Method | Accuracy (Omitted Obj.) | &nbsp; Probe Probability (Omitted Obj.) |
> | :--- | :---: | :---: |
> | **FLUX** | 0 | 0.292 |
> | **OSI - Omitted** | 0.70 | 0.658 |
> | **OSI - Present** | 0.14 | 0.298 |
>
> As shown in the table, when OSI is applied to the omitted object token, the accuracy of the omitted object reaches 0.70, and the probe probability increases from 0.292 to 0.658. In contrast, when the intervention is applied to the already present tokens, the increases in both probability and accuracy are marginal. This contrast demonstrates that the effectiveness of OSI relies on being applied to the specific target token. In conclusion, while our learned direction carries some general steering benefits, it also acts by directly amplifying the specific token signal to compel concept generation. We appreciate the reviewer's feedback that motivated this experiment, and we will include these results in the final paper.
>
> > **W2**: The label construction for probing is not fully clean. Using labels from final generated images for internal representations from intermediate timesteps means the supervision is only approximately aligned.
>
> To address your concern regarding the alignment between final labels and intermediate features, we conducted an additional experiment. At each denoising step, we extracted the one-step predicted clean image ($\hat{x}_0$) and assigned per-step presence labels using Mask2Former and BLIP-VQA, identical to the labeling pipeline used in our main paper. We then measured the agreement between these per-step $\hat{x}_0$ labels and the final image labels used in our main experiments. Using the same timestep thresholds defined in Figure 2 (a), we categorized the generation process into early, intermediate (the segment used for training our probe), and late timesteps, and averaged the agreement rates within each phase. The results are summarized below:
>
> | Timesteps | Agreement with Final Label |
> | :--- | :---: |
> | **Early** | 0.409 |
> | **Intermediate (Probe Training)** | 0.965 |
> | **Late** | 1.000 |
>
> In early timesteps, the low label agreement indicates that object presence remains undetermined during this stage. However, by the intermediate stage, the label agreement already reaches a highly agreement of 0.965. While the agreement reaches 1.000 in the late timesteps, these stages no longer capture sufficient information regarding concept omission, as previously analyzed in Figure 2. Therefore, our results demonstrate that using final image labels as supervision for intermediate features is well-justified and provides a highly aligned training signal.
>
> > **W3**: It is necessary to conduct a complete discussion of the ability boundaries and failure modes of this method.
>
> We appreciate the insightful feedback. To stay within space constraints, we have provided a detailed discussion of failure modes and boundaries in our response to **Reviewer TiFk** under the **Limitations** section. This includes an anonymous external link with visual examples. We commit to adding this discussion to the final paper. Please refer to that response for the full discussion.

---

> > ### Author Rebuttal · Reviewer_S7Fs · 2026-04-03
> >
> > Thanks for the author's response, my concern has been mostly resolved. I suggest the author include relevant analysis results and visualizations in the final version to enhance the readers' confidence.

---

> > > ### Author Response · Authors · 2026-04-03
> > >
> > > We are glad to hear that our response has mostly resolved your concerns. As you suggested, we will ensure that the relevant analysis results and visualizations are included in the final version. If there is any further information or clarification required for you to consider adjusting the score, please let us know.

---

### Decision · Program_Chairs · 2026-04-30

**Decision:**

Accept (regular)

**Comment:**

All three reviewers maintained positive recommendations for this paper. The reviewers consistently highlighted several strengths, including the thorough analysis of the problem, the effectiveness of the proposed method, and the convincing experimental results.

The main concerns centered on some overly strong claims in the presentation and the need for a clearer discussion of the method’s limitations. In the rebuttal, the authors addressed these issues to a satisfactory extent. Therefore, I am happy to recommend acceptance of this paper.

That said, the reviewers’ support for acceptance is contingent on the authors following through on their commitment to revise the relevant claims and discussion in the final version. I strongly encourage the authors to make these revisions carefully to ensure that the final paper reflects the scope and limitations of the work.